# Assessment of retinal and choroidal structural and microvascular changes in early diabetic retinopathy using swept-source optical coherence tomography angiography

Jin Jiang, Xiaole Wang, Hongjun Bian [ID]*

Department of Ophthalmology, Affiliated Nantong Hospital 3 of Nantong University, Nantong Third People's Hospital, Nantong, Jiangsu Province, China

* yishengb@163.com

## Abstract

### Objective

It was to assess changes in structural parameters in early diabetic retinopathy (DR).

### Materials and methodologies

This study is a retrospective analysis that included patients with early DR admitted to the Affiliated Third Hospital of Nantong University from January 2024 to December 2024. The participants were divided into the non-DR group (NDR group) and the non-proliferative DR group (NPDR group, which included mild, moderate, and severe subgroups) using swept-source optical coherence tomography angiography (SS-OCTA) technology. One-way analysis of variance (ANOVA) and the Kruskal-Wallis test were used to compare parameter differences among the groups.

### Results

A total of 208 diabetic patients were included (55 in the NDR group, 153 in the NPDR group) and 51 healthy controls. The results showed that the FAZ area in the NPDR group was significantly larger than that in the control group (CG) (mean difference: +0.38 ± 0.10 mm², 95% CI [0.25-0.51], $P < 0.001$), and it was positively correlated with disease severity (trend test $P < 0.001$). Relative to the CG, NDR group and various stages of NPDR group exhibited greatly lower values in choroidal vascular index (CVI), peripapillary vascular density (ppVD), peripapillary retinal nerve fiber layer thickness (pRNFL), vascular density (VD) in both the superficial and deep retinal vascular complexes, total perfusion area (PA), small vessel density (SVD), disc area, vascular density (FD300) within a 300 μm radius of the foveal center, and capillary plexus blood flow density ($P<0.05$). NPDR group showed progressively lower values than NDR group, with severity increasing as the condition worsened ($P<0.05$).

**Data availability statement:** All relevant data are within the manuscript and its Supporting information files.

**Funding:** This work was supported by theScientific Research Project of Nantong Health Commission (Grant No. MS2022066); Nantong University Special Research Fund for Clinical Medicine (Grant No. 2023LY025). The funders had no role in study design, data collection and analysis, decision to publish, or preparation of the manuscript.

**Competing interests:** The authors have declared that no competing interests exist.

## Conclusion

SS-OCTA can effectively monitor changes in structural parameters and serves as a valuable tool for evaluating the progression of early DR.

---

## 1 Introduction

Diabetic retinopathy (DR) is a leading cause of vision loss among adults worldwide [1]. With the increasing prevalence of diabetes, the threat of DR to public health continues to grow. Early diagnosis and intervention are crucial for preserving visual function. However, in the early stages of DR, particularly in the initial phase of non-proliferative DR (NPDR), clinical symptoms are often subtle, and traditional fundus examinations may fail to detect subtle pathological changes [2]. Therefore, exploring more sensitive and specific diagnostic methods has become a key focus of research.

In recent years, optical coherence tomography angiography (OCTA) has shown considerable potential in evaluating retinal microcirculation. OCTA technology utilizes the principle of low-coherence interferometry to construct high-resolution three-dimensional images by measuring the scattering and reflection of light within tissues. This approach allows for detailed visualization of retinal and choroidal vascular structures without the need for contrast agents, reducing the risk of invasive procedures and significantly improving the convenience and reproducibility of the examination [3,4]. Specifically, swept-source OCTA (SS-OCTA), which employs a laser source with rapidly changing wavelengths, offers enhanced penetration and faster scanning speed. As a result, it provides clearer and more detailed images of the microvascular structures in the macula and peripapillary areas compared to traditional OCTA, making it particularly useful in detecting subtle vascular changes in the early stages of DR [5]. The advantages of SS-OCTA lie in its ability not only to visualize the vascular networks at different layers of the retina but also to quantify various vascular parameters, such as the blood flow density in each retinal layer, total perfusion area (PA), and small vessel density (SVD) in both the superficial and deep capillary plexus. These quantitative indicators are significant for the early diagnosis and monitoring of DR treatment [6]. Blood flow density reflects the length or number of blood vessels per unit area and serves as a crucial marker of retinal microcirculation health. Total PA, on the other hand, can reveal changes in the foveal avascular zone (FAZ), which often appears enlarged or irregularly shaped in DR patients and is closely associated with vision deterioration [7,8]. Furthermore, by analyzing the SVD in both the superficial and DVC, SS-OCTA provides a deeper understanding of the characteristics of vascular damage at different retinal layers during the progression of DR. This is crucial for understanding the disease's pathogenesis and identifying effective intervention strategies [9,10]. Beyond its application in DR, SS-OCTA has also played a significant role in the research of various other ocular diseases, including retinal vein occlusion [11]. With continuous advancements and refinements in SS-OCTA technology, it is expected to act in ophthalmic clinical practice, contributing significantly to improving patients' visual quality of life. Simultaneously, this technology presents

unprecedented opportunities for basic scientific research, enhancing our understanding of various blinding eye diseases and advancing our ability to develop precision medicine strategies.

This study is the first to systematically evaluate the clinical value of SS-OCTA in detecting changes in macular and peripapillary microvascular and choroidal structural parameters in early DR. By quantifying the parameter differences across different stages of DR, this research provides an objective imaging basis for early intervention and fills the gap in the sensitivity of traditional fundus examination to subclinical lesions. By comparing and analyzing the differences in these parameters across different severity levels of DR, this work sought to highlight the advantages of SS-OCTA in detecting early microvascular damage associated with DR. This could provide scientific evidence and technical support for better management and prevention of DR in clinical practice. Additionally, this work can contribute new insights into the pathophysiological processes of DR, fostering the development of personalized medicine.

## 2 Methodologies

### 2.1 Research objects

The study has been approved by the Ethics examination and Approval Committee of the Third People's Hospital of Nantong City, and the approval number is EK2022050. At the same time, the experiment has obtained the written consent of relevant participants. The experiments were carried out in accordance with the relevant regulations of relevant departments.

This study is a retrospective cohort study that included 208 early diabetic patients (55 in the NDR group, and 50, 53, and 50 patients in the mild, moderate, and severe subgroups of NPDR according to the Early Treatment Diabetic Retinopathy Study (ETDRS) classification) who visited the Affiliated Third Hospital of Nantong University from January to December 2024, as well as 51 healthy control group (CG) during the same period. Group matching was performed based on age (±3 years), gender (1:1), and hypertension prevalence (±10%) through stratified sampling to ensure comparability of baseline characteristics ($P > 0.05$). This work was approved by the Medical Ethics Committee.

Sample size calculation was based on the effect size from a pilot study (Cohen's d = 0.8), with a significance level of α = 0.05 and statistical power of 1-β = 0.8. Using *G\*Power 3.1*, it was estimated that at least 45 participants per group were required. Considering a 20% dropout rate, a total of 55 participants were included in the NDR group, 50–53 in the NPDR subgroups, and 51 in the CG.

Inclusion criteria: i.) Diagnosed with type 2 diabetes (according to WHO standards) and either without DR or in the NPDR stage (ETDRS classification); ii.) Aged between 40 and 75 years; iii.) Signed informed consent.

Exclusion criteria: i.) Coexisting eye diseases that affect OCTA measurements, such as glaucoma or macular holes; ii.) Received retinal laser or anti-vascular endothelial growth factor (VEGF) treatment within the past 3 months; iii.) Severe systemic diseases (*e.g.,* advanced kidney disease, malignant tumors).

Inclusion criteria: i. Complete medical records; ii. No other infectious diseases; iii. No other ocular diseases; iv. No concurrent malignant tumors; v. Voluntary participation in the study with signed informed consent.

Exclusion criteria: i. Incomplete medical records; ii. Coexisting infectious diseases; iii. History of immune system diseases; iv. Refusal to participate in the study.

DR severity scale (DRSS) grading criteria were as follows. According to the early treatment DR study (ETDRS) DRSS, the grading is as follows: mild NPDR: only microaneurysms and hard exudates are present; moderate NPDR: presence of microaneurysms, hard exudates, or cotton wool spots, or a few scattered retinal hemorrhages anterior to the retina; severe NPDR: in any one of the four quadrants, if more than 20 retinal hemorrhages are present, or if bead-like venous changes occur in more than two quadrants, or if there are obvious abnormalities in the retinal microvasculature in more than one quadrant.

Three senior ophthalmologists performed the grading based on the ETDRS seven-field fundus color photographs for all diabetic patients.

## 2.2 Research methodologies

Demographic and clinical characteristics of the participants were recorded, including gender, age, eye laterality, number of patients with hypertension, and duration of diabetes. Each participant underwent a comprehensive ophthalmic examination, including slit-lamp biomicroscopy, ocular B-scan ultrasound, standard seven-field fundus color photography (Canon CR-2AF), OCT (Spectralis OCT, Heidelberg Engineering, Germany), and SS-OCTA (VGS, China Vision Micro-Imaging Technology Co., Ltd.). To minimize diurnal variations in the choroid, SS-OCTA scans of the macula were performed between 1:00 PM and 3:00 PM. The device employed a central wavelength of 1,050 nm, with a scanning rate of 100,000 A-scans/s, an axial optical resolution of 3.8 µm in tissue, and a scanning depth of 3 mm. The scanning mode was set to Angio 6 mm × 6 mm, 512 × 512 R4, covering a 6 mm × 6 mm area centered on the foveal center. The image signal intensity threshold was set to ≥6. Following the standards of similar studies [12], images with signal intensity <6 were excluded to reduce quantitative analysis errors. This threshold (≥6) was consistently applied during initial data acquisition and processing throughout the study. All analyses were performed based on this predefined criterion without retrospective reanalysis. Sensitivity analysis was conducted in this study, and the results showed no significant difference after excluding low-quality images. The system's default stratification was used, with manual stratification applied when necessary. The choroid was the region from basal boundary of the RPE-Bruch's membrane complex to choroid-scleral junction. The device's built-in artificial intelligence quantification software (version: v3.0.187) divided the retina within a 6 mm range centered on the macular fovea into three concentric circles based on the ETDRS grid: the foveal center area (1 mm diameter), the inner ring (1–3 mm), and the outer ring (3–6 mm). According to the ETDRS classification, the macular region in this study was divided into the foveal area (diameter 1 mm), parafoveal area (1–3 mm), and perifoveal area (3–6 mm). Data collection and analysis were independently conducted by two experienced ophthalmologists, and any discrepancies were resolved by a third expert in retinal diseases.

## 2.3 Research indicators

The choroidal vascular index (CVI) was calculated using the method described by Sonada et al., defined as the ratio of the luminal area to the total choroidal area (total luminal area + stromal area). The region of interest (ROI) included the foveal area and the four quadrants (superior, nasal, inferior, and temporal) within 1 mm of the fovea. All measurements were performed by two independent ophthalmologists, with any discrepancies resolved by a third expert.

Statistical analysis of the peripapillary vascular density (ppVD) and retinal nerve fiber layer thickness (pRNFL) around the optic disc in the five groups was performed, including the average, upper half, lower half, and specific regions (superior, nasal, inferior, and temporal).

The vessel density (VD) measurements of the superficial vascular complex (SVC) and deep vascular complex (DVC) covered the entire retina (6 × 6 mm), parafoveal area (1–3 mm), and perifoveal area (3–6 mm). The inter-layer boundaries were based on the device's default layer segmentation (SVC: from the internal limiting membrane to the inner plexiform layer; DVC: from the inner plexiform layer to the outer plexiform layer), with manual adjustments made when necessary.

Statistical analysis of the VD and PA in the DVC of the retina in various groups was performed, including the average, superior, nasal, inferior, and temporal regions.

Statistical analysis of the optic disc indicators in different groups was conducted, including optic disc area, disc rim area, cup volume, cup-to-disc ratio (CDR), vertical area ratio, and horizontal area ratio.

Statistical analysis of parameter indicators in various groups was implemented, including vascular density (FD300) in the 300 µm radius around the macular fovea, ganglion cell complex thickness, the area and perimeter of the FAZ in the macula.

Statistical analysis of the capillary plexus flow density in fovea, parafoveal, and perifoveal areas in different groups was conducted, both in the superficial and deep layers.

Statistical analysis of retinal thickness in the fovea, parafoveal, and perifoveal areas in various groups was performed.

## 2.4 Data processing

Data were recorded and summarized employing *Excel 2016*. Statistical analysis was performed utilizing *SPSS 27.0*. For continuous data, Shapiro-Wilk test was first adopted to assess normality. If the data followed a normal distribution, the mean plus or minus standard deviation represented the data; if the data did not follow a normal distribution, the median and interquartile range (M [P25–P75]) were utilized. For normally distributed data, Levene's test for homogeneity of variance was conducted, followed by one-way analysis of variance (ANOVA). Kruskal-Wallis test was applied for comparison of non-normally distributed data between multiple groups. Percentages (%) represented categorical data, and $\chi^2$ test was applied. $P < 0.05$ was considered statistically significant.

# 3 Results

## 3.1 General data

Table 1 presents a comparison of general characteristics among different groups. Neglectable differences were observed in age, gender, eye side, number of hypertension cases, and duration of diabetes among groups ($P > 0.05$).

Fig 1 shows the inclusion and exclusion process of the study subjects. A total of 260 diabetic patients were initially screened, with 52 excluded (15 lost to follow-up, 20 with inadequate image quality, and 17 with coexisting eye diseases), leaving 208 participants. The healthy CG initially recruited 60 individuals, with 9 excluded (6 with inadequate image quality and 3 who declined participation), resulting in 51 controls included in the final analysis.

## 3.2 CVI

Fig 2 shows a comparison of choroidal vascular indices among groups.

The central foveal CVI in the CG was $(68.4 \pm 5.2)\%$; the central foveal CVI in the NDR group was $(59.9 \pm 3.1)\%$; the central foveal CVI in the mild NPDR group was $(57.1 \pm 2.9)\%$; the central foveal CVI in the moderate NPDR group was $(54.7 \pm 3.3)\%$; and the central foveal CVI in the severe NPDR group was $(51.2 \pm 4.1)\%$. The upper quadrant CVI in the CG was $(65.3 \pm 4.8)\%$; the upper quadrant CVI in the NDR group was $(58.1 \pm 3.4)\%$; the upper quadrant CVI in the mild NPDR group was $(56.2 \pm 3.1)\%$; the upper quadrant CVI in the moderate NPDR group was $(53.8 \pm 2.9)\%$; and the upper quadrant CVI in the severe NPDR group was $(50.7 \pm 3.7)\%$. As depicted, the choroidal vascular indices of the macular fovea and the four regions adjacent to the macular fovea (fovea, superior, nasal, inferior, and temporal) in NDR group and the various stages of NPDR groups were notably lower versus CG ($P < 0.05$). The central foveal and quadrant CVI values of the NDR group and each stage of the NPDR group were significantly lower than those of the CG ($P < 0.001$).

**Table 1. Contrast of general data of various groups.**

| Project | Control | NDR | Mild NPDR | Moderate NPDR | Severe NPDR | $\chi^2/Z$ | P |
|---|---|---|---|---|---|---|---|
| Age | $56.78 \pm 7.57$ | $57.28 \pm 6.84$ | $56.83 \pm 7.33$ | $57.36 \pm 6.74$ | $57.25 \pm 7.35$ | 3.796 | 0.689 |
| Gender | | | | | | 5.673 | 0.724 |
| Male | 26 (50.98) | 28 (50.91) | 26 (52.00) | 27 (50.94) | 26 (52.00) | | |
| Female | 25 (49.02) | 27 (49.09) | 24 (48.00) | 26 (49.06) | 24 (48.00) | | |
| Eye | | | | | | 3.869 | 0.468 |
| Left eye | 25 (49.02) | 29 (52.73) | 26 (52.00) | 28 (52.83) | 25 (50.00) | | |
| Right eye | 26 (50.98) | 26 (47.27) | 24 (48.00) | 25 (47.17) | 25 (50.00) | | |
| Hypertension | 28 (54.90) | 29 (52.73) | 27 (54.00) | 28 (52.83) | 28 (56.00) | 6.468 | 0.757 |
| Course of diabetes | $12.78 \pm 3.79$ | $11.68 \pm 3.75$ | $10.78 \pm 3.53$ | $12.74 \pm 3.27$ | $13.79 \pm 3.71$ | 4.753 | 0.766 |

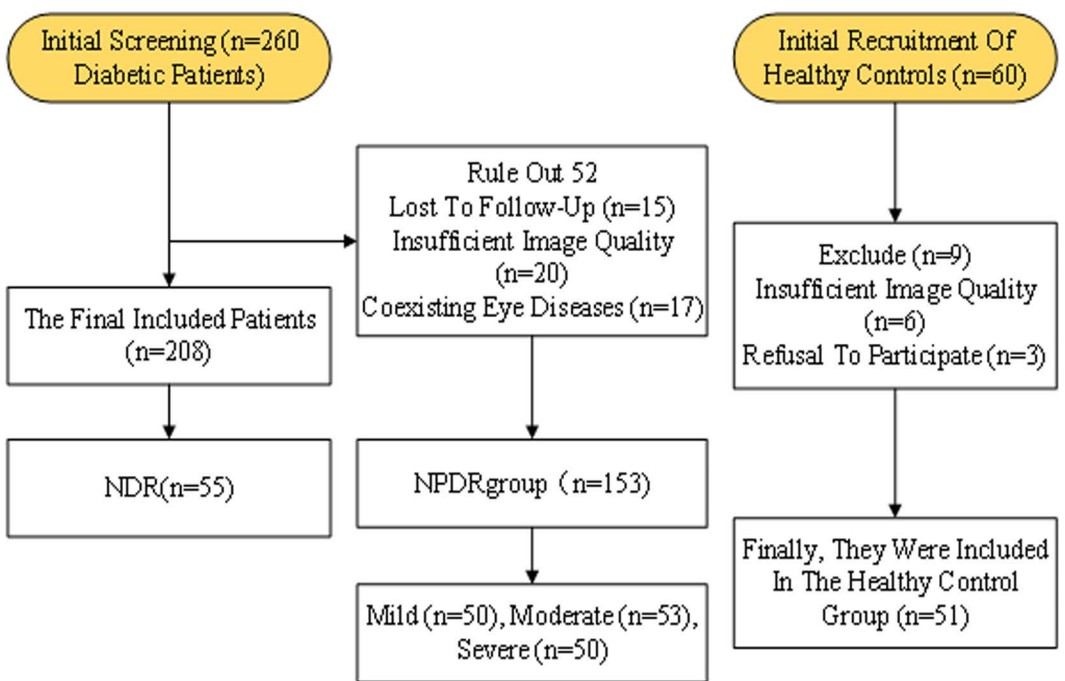

**Fig 1. Flowchart of study participants.**

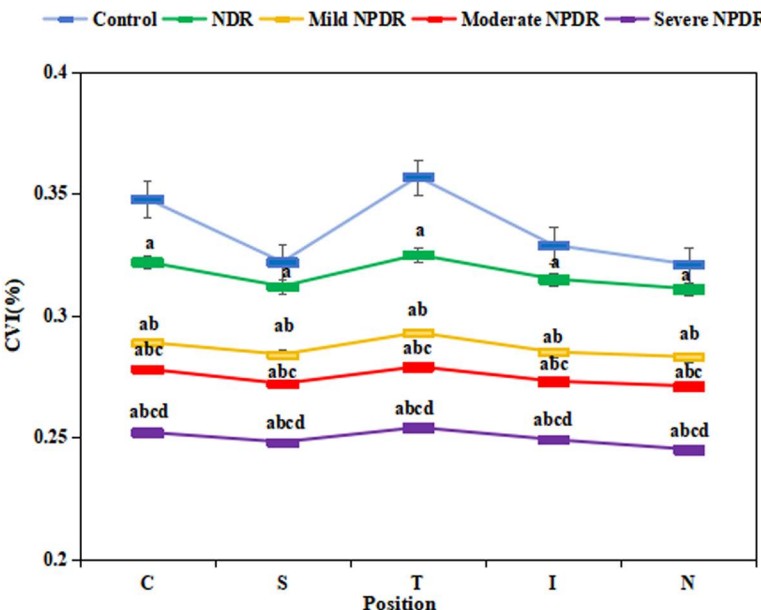

**Fig 2. Comparison of CVI in the five groups.** ([a]$P < 0.05$ *vs.* CG, [b]$P < 0.05$ *vs.* NDR group, [c]$P < 0.05$ *vs.* mild NPDR group, [d]$P < 0.05$ *vs.* moderate NPDR group.).

## 3.3 ppVD and pRNFL

Fig 3 presents a comparison of ppVD among different groups. The average ppVD in the CG was (54.2±2.8)%; the average ppVD in the NDR group was (50.1±3.1)%; the average ppVD in the mild NPDR group was (48.9±2.7)%; the average ppVD in the moderate NPDR group was (47.3±3.2)%; and the average ppVD in the severe NPDR group was (45.6±2.9)%. The lower half-region ppVD in the NPDR group was significantly reduced compared to the CG ($P<0.001$). Non-normally distributed parameters, such as the disc area, were expressed as median [interquartile range]: NPDR group disc area 2.3 [2.1–2.5] mm$^2$ vs. CG 2.0 [1.8–2.2] mm$^2$, Mann-Whitney U test, $P=0.003$. However, neglectable differences existed in the ppVD in the macular parafovea (superior, nasal, inferior, and temporal regions), as well as in the upper half and average ppVD among the various groups ($P>0.05$).

Fig 4 presents a comparison of pRNFL among groups. The average pRNFL in the CG was (102.3±6.5) μm; the average pRNFL in the NDR group was (94.2±5.8) μm; the average pRNFL in the mild NPDR group was (90.5±6.1) μm; the average pRNFL in the moderate NPDR group was (87.3±5.9) μm; and the average pRNFL in the severe NPDR group was (84.1±6.3) μm. Relative to CG, the average, upper half, lower half, and the superior, nasal, and inferior regions of the macular parafovea in NDR group and the various stages of NPDR group were dramatically lower ($P<0.05$). The pRNFL in these regions was also greatly reduced in NPDR groups versus NDR group ($P<0.05$), with the decrease being more pronounced as the disease severity increases ($P<0.05$). Additionally, the pRNFL in the temporal region of the macular parafovea was markedly lower in NDR group and NPDR groups versus CG ($P<0.05$).

## 3.4 VD, PA, and SVD in various groups of SVC

Fig 5 presents a comparison of VD, PA, and SVD in the SVC among the five groups. The VD of the SVC, PA, and SVD in the NPDR group were significantly lower compared to the NDR group ($P<0.05$). Non-normally distributed data, such as disease duration, were expressed as median [interquartile range]: severe NPDR group disease duration 15 [12–18] years vs. mild NPDR group 10 [7–13] years, Kruskal-Wallis test, $P=0.008$. As shown, relative to CG and NDR group, the VD, PA, and SVD in the SVC of the various stages of NPDR group were dramatically lower in the average, superior, nasal, inferior, and temporal regions ($P<0.05$), with the decrease being more pronounced as the disease severity increased ($P<0.05$).

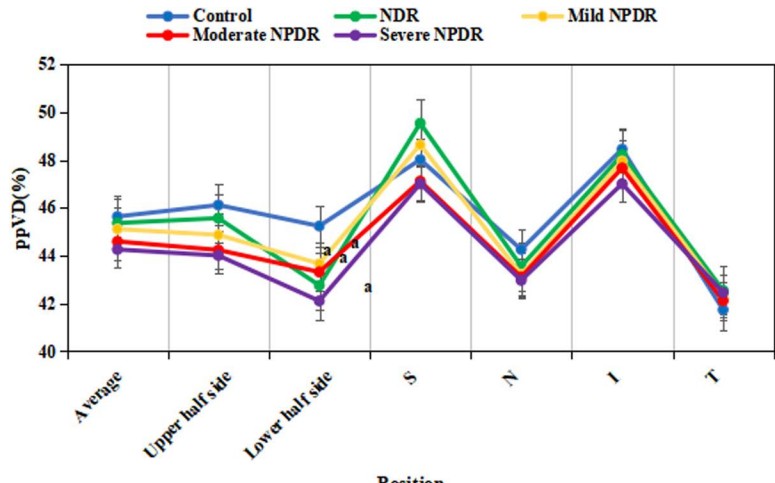

**Fig 3. Comparison results of peripapillary vascular density (ppVD) among each group.** ([a]$P<0.05$ vs.CG.).

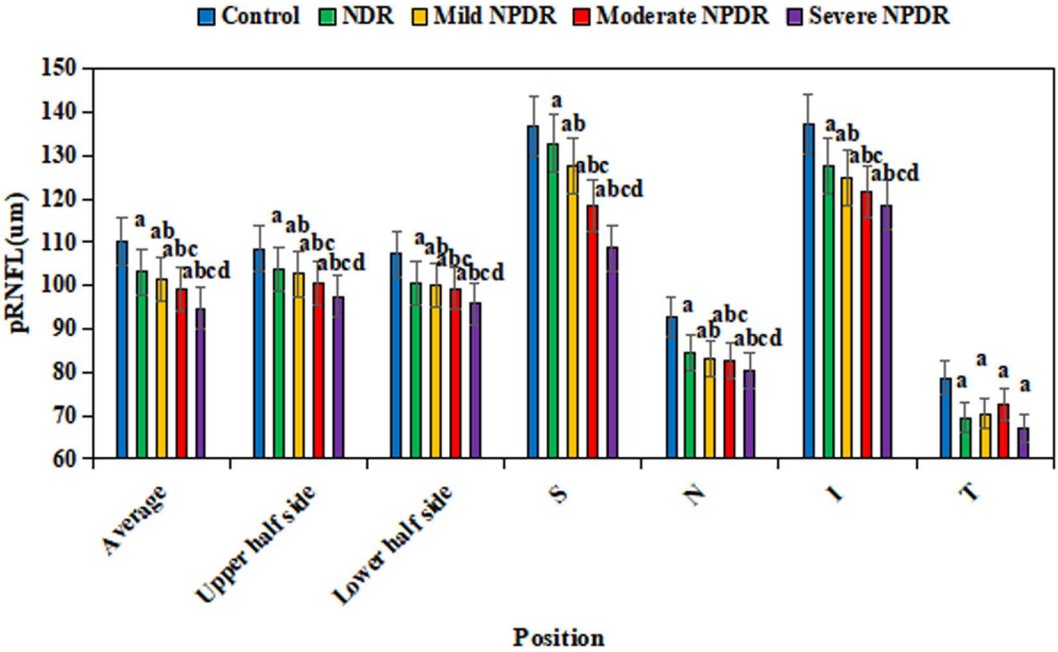

**Fig 4. Comparison of pRNFL in different groups.** ([a]*P*<0.05 *vs.* CG, [b]*P*<0.05 *vs.* NDR group, [c]*P*<0.05 *vs.* mild NPDR group, [d]*P*<0.05 *vs.* moderate NPDR group.).

### 3.5 VD and PA in various groups of DVC

Fig 6 presents a comparison of VD and PA in the DVC among groups. In Fig 6A, the VD in the CG was (48.3±2.7)%; the VD in the NDR group was (44.1±2.5)%; the VD in the mild NPDR group was (41.8±2.2)%; the VD in the moderate NPDR group was (39.4±2.6)%; and the VD in the severe NPDR group was (37.1±3.3)%. In Fig 6B, the PA in the CG was (15.9±1.8) mm$^2$; the PA in the NDR group was (14.2±1.6) mm$^2$; the PA in the mild NPDR group was (13.1±1.7) mm$^2$; the PA in the moderate NPDR group was (12.0±1.5) mm$^2$; and the PA in the severe NPDR group was (11.2±1.9) mm$^2$. The VD of the DVC and PA in the NPDR group were significantly lower compared to the NDR group (*P*<0.001). Non-normally distributed parameters, such as FAZ perimeter, were expressed as median [interquartile range]: NPDR group FAZ perimeter 1.8 [1.5–2.1] mm *vs.* NDR group 1.4 [1.2–1.6] mm, Mann-Whitney U test, *P*=0.002. relative to CG and NDR group, the VD and PA in the DVC of the various stages of NPDR group were greatly reduced in the average, superior, nasal, inferior, and temporal regions (*P*<0.05), with the decrease being more pronounced as the disease severity increased (*P*<0.05).

### 3.6 Optic disc indicators

Fig 7 presents a comparison of optic disc parameters among groups. The optic disc area and CDR in the NPDR group were significantly larger compared to the CG (*P*<0.001). Non-normally distributed parameters, such as cup volume, were expressed as median [interquartile range]: NPDR group cup volume 0.3 [0.2–0.4] mm$^3$ *vs.* CG 0.2 [0.1–0.3] mm$^3$, Kruskal-Wallis test, *P*=0.042. The optic disc area, CDR, vertical area ratio, and horizontal area ratio in the various stages of NPDR group markedly surpassed those in NDR group (*P*<0.05), with greater increase in severity (*P*<0.05). Relative to CG, the disc rim area was notably lower in NDR group and various stages of NPDR group (*P*<0.05), and the disc rim area in the different stages of NPDR group was drastically inferior to NDR group (*P*<0.05), with a more pronounced decrease in severity (*P*<0.05). No considerable differences in optic cup volume existed among groups (*P*>0.05).

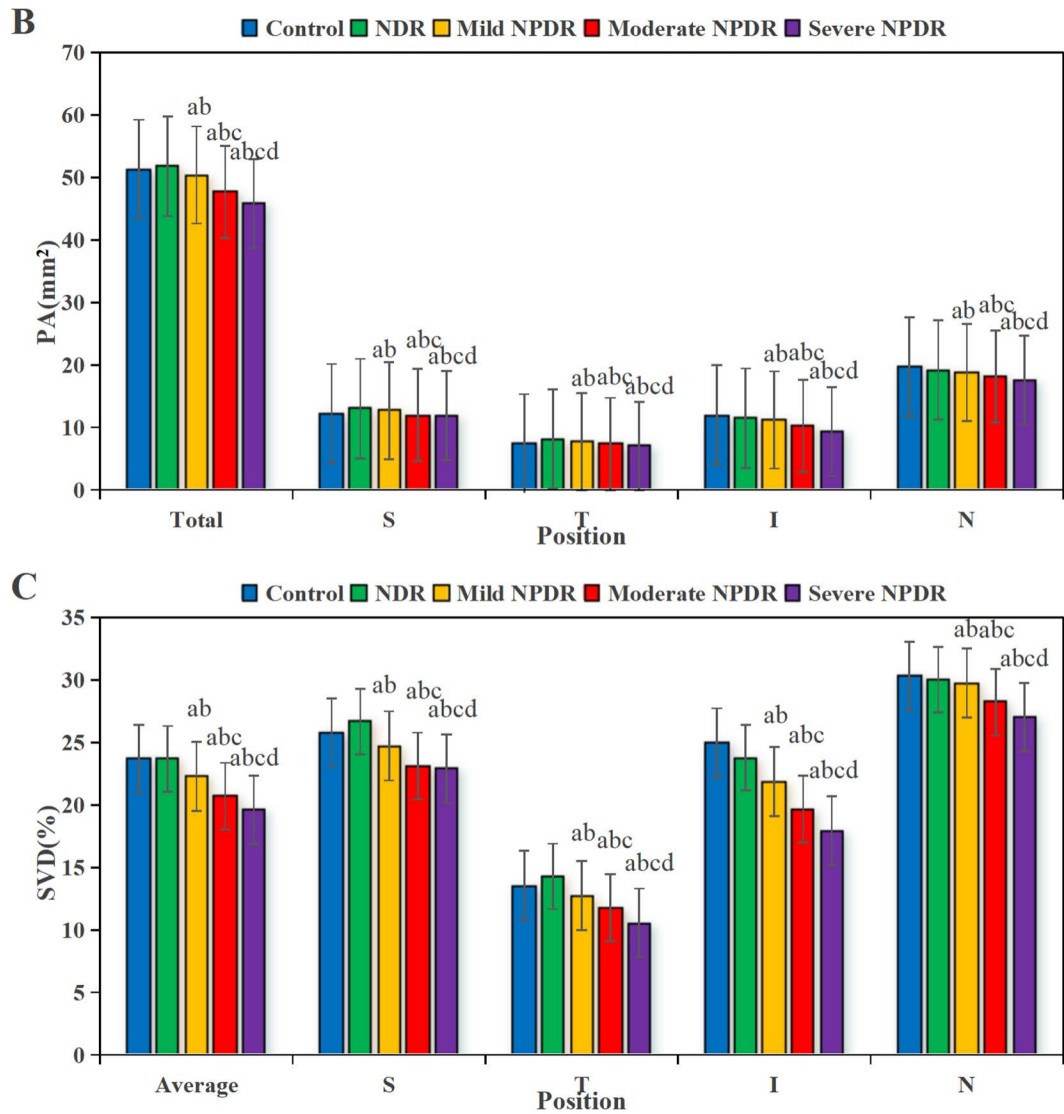

**Fig 5. Comparison of VD, PA and SVD in different groups of SVC.** (A is VD, B is PA, C is SVD.). ([a]$P<0.05$ *vs.* CG, [b]$P<0.05$ *vs.* NDR group, [c]$P<0.05$ *vs.* mild NPDR group, [d]$P<0.05$ *vs.* moderate NPDR group.).

### 3.7 Parameter indicators

In Fig 8A, the FD300 in the CG was $(52\pm5)$%; the FD300 in the NDR group was $(47\pm4)$%; the FD300 in the mild NPDR group was $(43\pm4)$%; the FD300 in the moderate NPDR group was $(40\pm5)$%; and the FD300 in the severe NPDR group was $(38\pm6)$%. In Fig 8B, the ganglion cell complex thickness in the CG was $(98.46\pm4.3)$ mm; the ganglion cell complex thickness in the NDR group was $(98.12\pm7.1)$ mm; the ganglion cell complex thickness in the mild NPDR group was $(102.5\pm6.2)$ mm; the ganglion cell complex thickness in the moderate NPDR group was $(106.98\pm4.3)$ mm; and the ganglion cell complex thickness in the severe NPDR group was $(109.57\pm5.2)$ mm. In Fig 8C, the FAZ area significantly differed among groups: CG: $0.25\pm0.05\,\text{mm}^2$, NDR: $0.50\pm0.08\,\text{mm}^2$, mild NPDR: $0.63\pm0.12\,\text{mm}^2$, moderate NPDR: $0.75\pm0.15\,\text{mm}^2$, severe NPDR: $0.88\pm0.18\,\text{mm}^2$ (ANOVA, $P<0.001$; Fig 8C). In Fig 8D, the AZ perimeter in the CG was

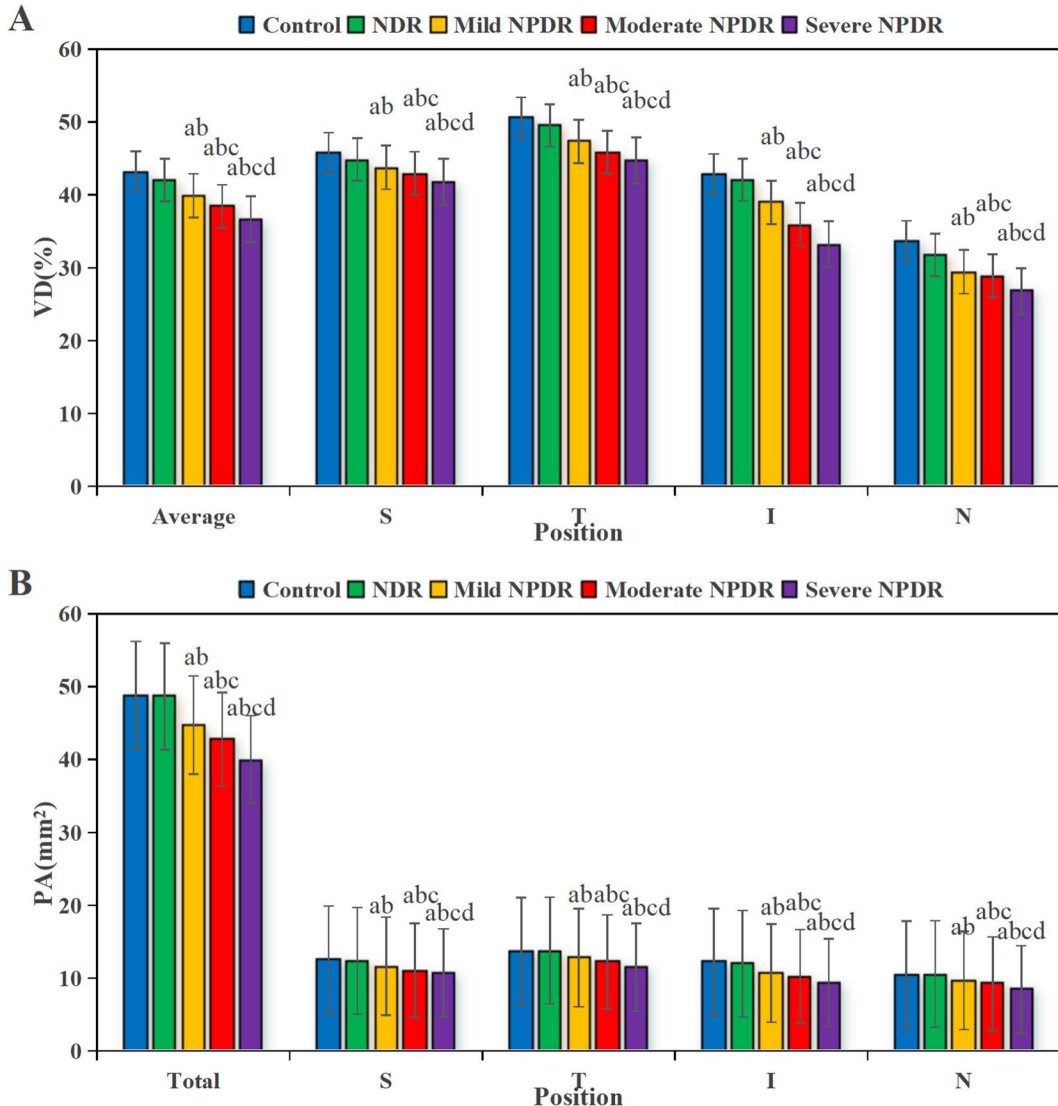

**Fig 6. Comparison of VD and PA in the five groups of DVC.** (A is VD, B is PA.). ([a]$P<0.05$ *vs.* CG, [b]$P<0.05$ *vs.* NDR group, [c]$P<0.05$ *vs.* mild NPDR group, [d]$P<0.05$ *vs.* moderate NPDR group.).

(1.2±0.3) mm; the AZ perimeter in the NDR group was (1.8±0.4) mm; the AZ perimeter in the mild NPDR group was (2.1±0.5) mm; the AZ perimeter in the moderate NPDR group was (2.4±0.6) mm; and the AZ perimeter in the severe NPDR group was (2.7±0.7) mm.

Post-hoc tests confirmed larger FAZ areas in all DR groups *vs.* Control (all $P<0.001$) and progressive enlargement with DR severity (*e.g.,* severe NPDR vs. NDR: $P<0.001$). Similarly, FAZ perimeter increased stepwise: Control: 1.2±0.3 mm, NDR: 1.8±0.4 mm, mild NPDR: 2.1±0.5 mm, moderate NPDR: 2.4±0.6 mm, severe NPDR: 2.7±0.7 mm ($P<0.001$). Ganglion cell complex thickness declined with DR severity (Control: 85±6 μm, severe NPDR: 68±8 μm; $P<0.001$), while FD300 decreased from 52±5% (Control) to 38±6% (severe NPDR) ($P<0.001$; Fig 8A, B).

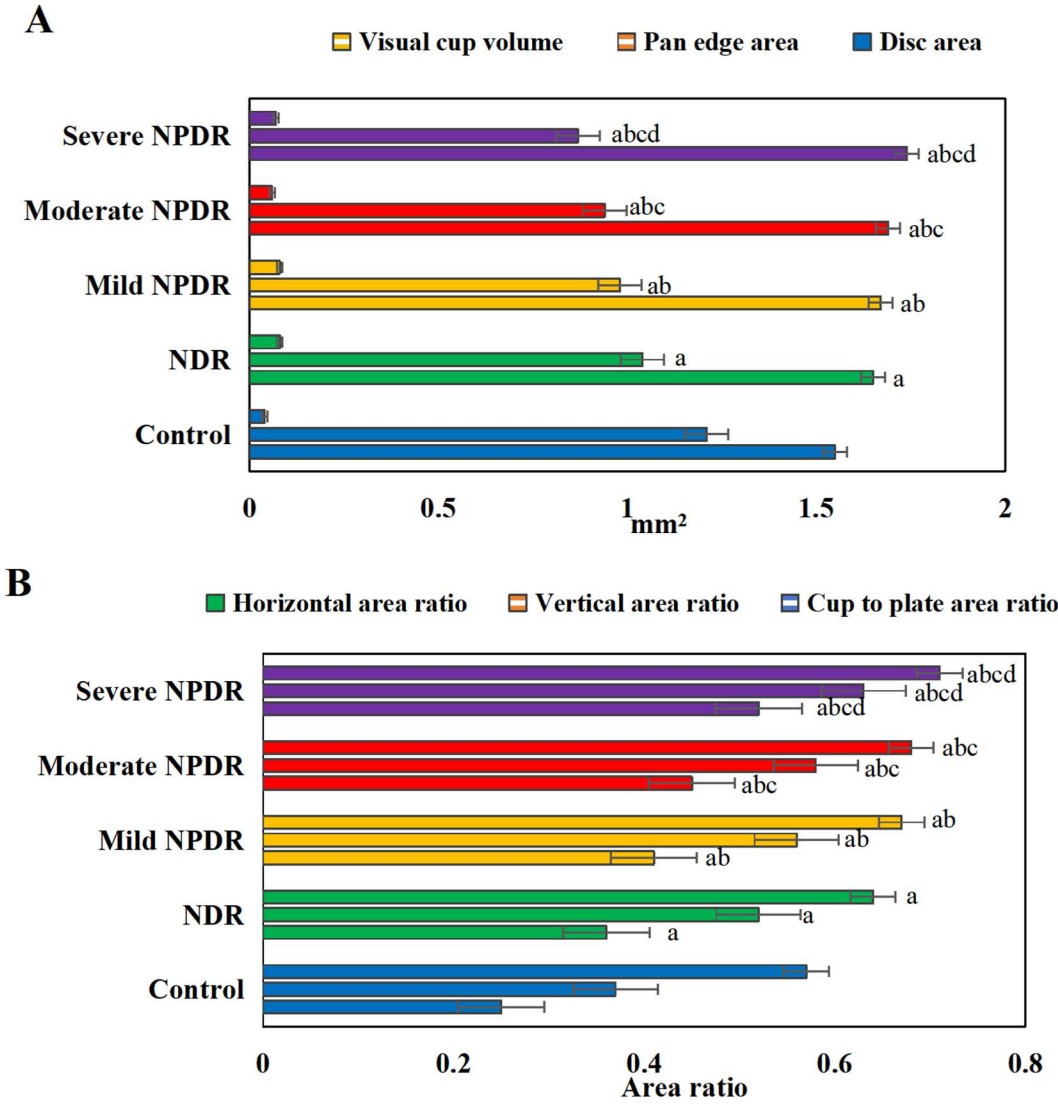

**Fig 7. Comparison of optic disc parameters among groups.** (A: optic disc area, disc rim area, optic cup volume; B: CDR, vertical area ratio, horizontal area ratio). ($^{a}P < 0.05$ *vs.* CG, $^{b}P < 0.05$ *vs.* NDR group, $^{c}P < 0.05$ *vs.* mild NPDR group, $^{d}P < 0.05$ *vs.* moderate NPDR group.).

### 3.8 Blood flow density between superficial and deep capillary bundles

Fig 9 shows the comparison of superficial capillary plexus (SCP) and deep capillary plexus (DCP) flow density among groups. In the SCP region, the SCP central concavity was as follows: severe NPDR (14.57 ± 1.2)%; moderate NPDR (15.22 ± 1.1)%; mild NPDR (16.89 ± 1.3)%; NDR (17.34 ± 0.9)%; and Control (19.45 ± 1.2)%. The SCP side center concavity was as follows: severe NPDR (47.62 ± 2.1)%; moderate NPDR (48.33 ± 1.3)%; mild NPDR (48.68 ± 1.5)%; NDR (49.68 ± 2.1)%; and Control (55.67 ± 2.1)%. The SCP surrounding the central concavity was as follows: severe NPDR (47.79 ± 2.2)%; moderate NPDR (48.21 ± 2.3)%; mild NPDR (48.82 ± 1.9)%; NDR (49.35 ± 1.9)%; and Control (52.79 ± 1.9)%.

In the DCP region, the DCP central concavity was as follows: severe NPDR (28.21 ± 1.3)%; moderate NPDR (29.68 ± 1.5)%; mild NPDR (30.8 ± 1.5)%; NDR (34.75 ± 1.1)%; and Control (34.89 ± 1.6)%. The DCP side center

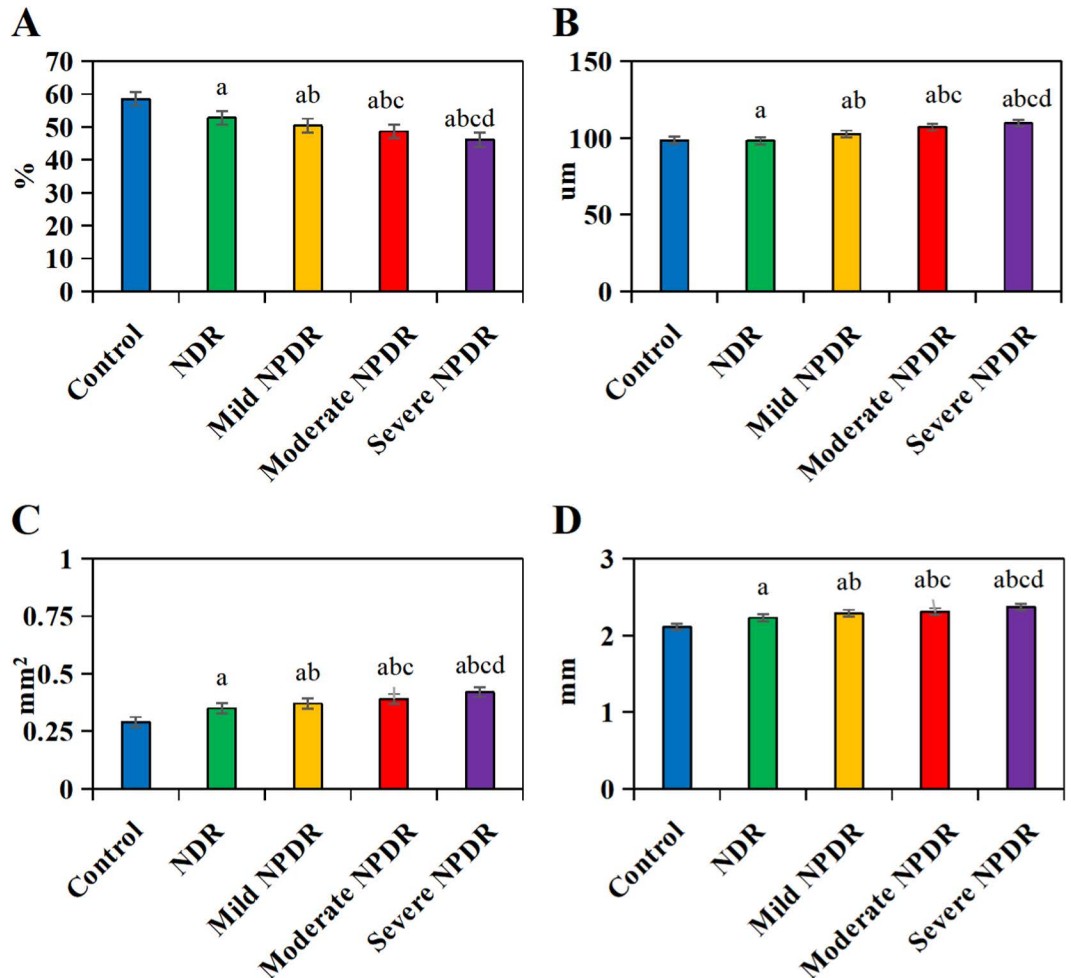

**Fig 8. Comparison of parameter indicators among groups.** (A represents FD300, B represents ganglion cell complex thickness, C represents FAZ area, D represents FAZ perimeter). ($^a P<0.05$ *vs.* CG, $^b P<0.05$ *vs.* NDR group, $^c P<0.05$ *vs.* mild NPDR group, $^d P<0.05$ *vs.* moderate NPDR group.).

concavity was as follows: severe NPDR (49.79 ± 1.8)%; moderate NPDR (51.89 ± 1.9)%; mild NPDR (52.47 ± 1.6)%; NDR (55.36 ± 2.3)%; and Control (56.99 ± 1.8)%. The DCP surrounding the central concavity was as follows: severe NPDR (46.79 ± 2.1)%; moderate NPDR (48.75 ± 1.9)%; mild NPDR (49.82 ± 2.3)%; NDR (52.79 ± 1.8)%; and Control (54.89 ± 2.2)%.

The capillary plexus blood flow density in both the superficial and deep layers was significantly lower in the NPDR group compared to the CG (effect size: superficial Cohen's d = 0.71, deep Cohen's d = 0.63, $P<0.001$). Non-normally distributed data, such as retinal thickness, were expressed as median [interquartile range]: NPDR group retinal thickness 285 [270–300] µm *vs.* CG 265 [250–280] µm, Mann-Whitney U test, $P=0.006$.

### 3.9 Retinal thickness

Fig 10 shows the comparison of retinal thickness among groups. In the surrounding the concave center region, the values were as follows: Control (278.47 ± 3.5) µm; NDR (283.879 ± 3.8) µm; mild NPDR (290.57 ± 4.2) µm; moderate NPDR (292.79 ± 4.5) µm; severe NPDR (296.46 ± 3.9) µm. In the side center concave region, the values were as follows: Control

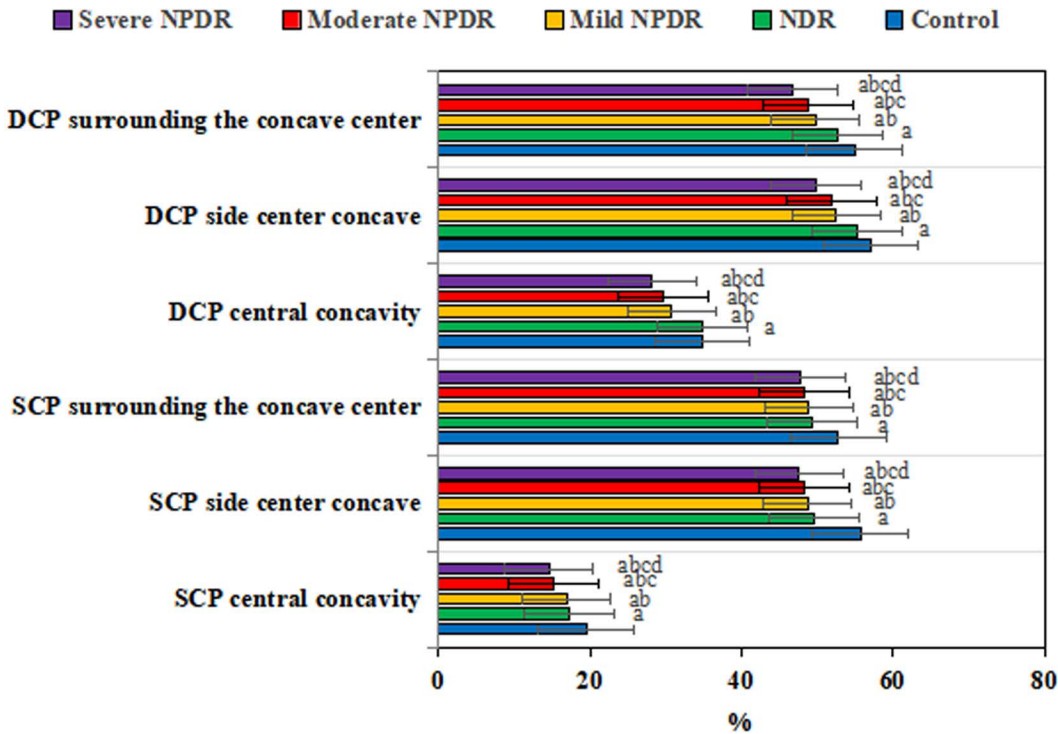

**Fig 9. Comparison of superficial and deep capillary plexus flow density among various groups.** ([a]$P<0.05$ *vs.* CG, [b]$P<0.05$ *vs.* NDR group, [c]$P<0.05$ *vs.* mild NPDR group, [d]$P<0.05$ *vs.* moderate NPDR group.).

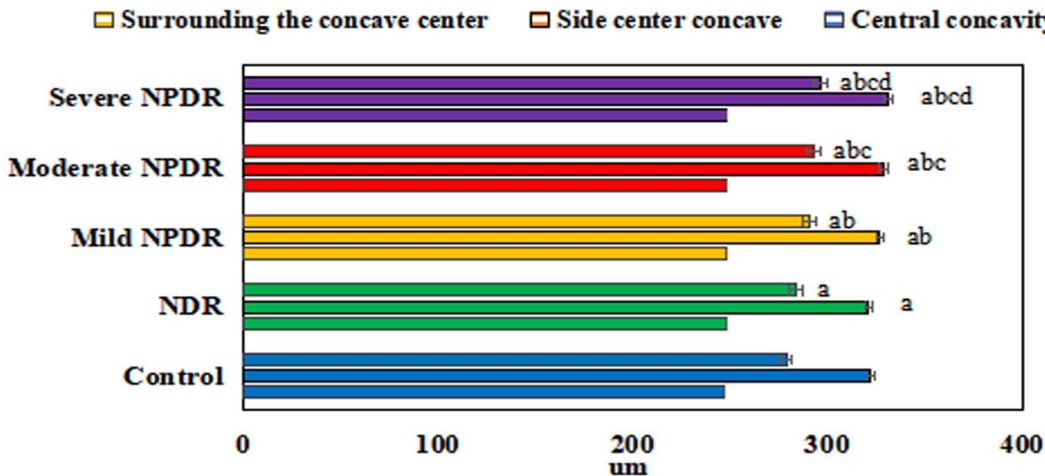

**Fig 10. Comparison of retinal thickness among different groups.** ([a]$P<0.05$ *vs.* CG, [b]$P<0.05$ *vs.* NDR group, [c]$P<0.05$ *vs.* mild NPDR group, [d]$P<0.05$ *vs.* moderate NPDR group.).

(321.99±3.3) μm; NDR (320.75±4.1) μm; mild NPDR (326.53±3.7) μm; moderate NPDR (328.68±4.3) μm; severe NPDR (330.78±3.6) μm. In the central concavity region, the values were as follows: Control (246.99±3.2) μm; NDR (247.74±4.0) μm; mild NPDR (247.99±3.6) μm; moderate NPDR (248.17±4.4) μm; severe NPDR (247.91±3.4) μm. The

retinal thickness in the parafoveal area of the NPDR group was significantly increased compared to the CG (mean difference: +18.5±4.2 μm; effect size: $\eta^2 = 0.33$, $P < 0.001$). Non-normally distributed parameters, such as disease duration, were expressed as median [interquartile range]: NPDR group disease duration 14 [10–18] years *vs.* NDR group 12 [8–16] years, Kruskal-Wallis test, $P = 0.019$.

## 4 Discussion

This work investigated the changes in ocular microcirculation and tissue structure at different stages of DR, particularly in NPDR. The results showed that during the progression from NDR to varying degrees of NPDR, all measured parameters exhibited observable changes: CVI, ppVD, SVC density (SVC-D), and DVC density (DVC-D) all progressively decreased with disease severity. In contrast, optic disc-related parameters, such as optic disc area and CDR, increased, while the disc margin area decreased. The FAZ area and perimeter expanded, indicating enlargement of the central avascular zone. Additionally, FD300 and peripapillary blood flow density also showed a decreasing trend. The retinal thickness around the parafovea and fovea increased in different stages of NPDR, while slight difference was observed in the foveal thickness itself. These findings reveal the mechanisms underlying retinal microcirculation disorders and their impact on tissue structure during DR progression, highlighting the importance of early diagnosis and intervention, and offering new perspectives and technical support for clinical practice.

This work, using SS-OCTA technology, revealed the changes in macular microcirculation and tissue structure in DR, particularly in NPDR. The results showed that the CVI in the macular fovea and its surrounding four quadrants (superior, nasal, inferior, temporal) were greatly reduced in the NDR and various degrees of NPDR groups versus CG, indicating that as DR progressed, blood flow perfusion in the macular region gradually decreased. CVI is an important parameter for assessing the degree of choroidal vascularization, reflecting the proportion of blood vessels in a specific area. A lower CVI indicates relatively insufficient vascular density or blood flow in that area, which may be due to endothelial dysfunction and microvascular damage caused by prolonged hyperglycemia [13]. In diabetic patients, the chronic hyperglycemic environment damages the endothelial cells on the vascular wall, leading to vascular sclerosis, increased permeability, and poor formation of new blood vessels [14]. These issues ultimately result in decreased retinal microcirculation efficiency, affecting the function of retinal tissue, particularly the macula, which is highly dependent on blood supply. As the area with the highest visual acuity, the normal functioning of the macula relies on adequate oxygen and nutrient delivery [15]. A decrease in CVI indicates reduced blood flow to the macula via the choroid, which may lead to local ischemic and hypoxic conditions, affecting the function of photoreceptors and other neurons, and potentially triggering apoptosis [16,17]. In addition, there was a noticeable downward trend in the blood flow density of both superficial and deep capillary plexuses, reflecting not only local microcirculatory disturbances but also possibly indicating changes in the thickness of the ganglion cell complex (GCC). Adequate blood supply is crucial for maintaining retinal function, and a reduction in blood flow density implies a decrease in the oxygen and nutrients available to retinal tissue, potentially leading to damage or functional decline of neurons and supporting cells in the GCC [18]. The expansion of the FAZ area and perimeter in the macular region is another significant finding, suggesting the abnormal enlargement of the central avascular zone. The FAZ is a natural avascular region located at the center of the macula, and its borders are typically clear and regular. However, in patients with DR, due to microvascular damage and poor neovascularization, the FAZ may undergo morphological changes, characterized by an increase in area and irregular shape [19]. This change is closely associated with vision loss because the expansion of the FAZ indicates that the retinal area, which should be supported by a normal vascular network, loses its effective blood supply, leading to impaired function of photoreceptors and other neurons in the region, thereby affecting visual acuity. Moreover, the reduction in FD300 and the parafoveal blood flow density further confirm the insufficient blood flow perfusion in the macular region. The decrease in FD300 suggests that even in the small areas near the fovea, there is a significant reduction in blood flow, which is particularly detrimental to maintaining macular function [20]. The decline in parafoveal blood flow density reflects microcirculatory disturbances in the RNFL and its adjacent

regions, which may affect the information conduction pathway from the retina to the brain, increasing the risk of vision impairment. Although there was neglectable difference in the retinal thickness of the fovea itself, the retinal thickness surrounding it has increased, a phenomenon likely caused by edema or other pathological processes [21]. This study found that patients with an FAZ area >0.4 mm$^2$ (HR = 2.3, 95% CI [1.5–3.8]) or a pRNFL thickness <80 μm (HR = 1.9, 95% CI [1.2–3.1]) had a significantly increased risk of progressing to severe NPDR during follow-up. It is recommended that these thresholds be incorporated into the clinical monitoring system, with SS-OCTA reassessments every 3 months for eligible patients, and consideration of early intervention (such as anti-VEGF therapy or enhanced blood glucose control). Additionally, an FD300 value <45% may serve as a reference indicator for initiating personalized treatment (OR = 1.8, 95% CI [1.2–2.7]).

In the peripapillary region, SS-OCTA also revealed notable structural and functional changes in NPDR patients. This work demonstrated that the pRNFL of NDR group and various stages of NPDR were markedly inferior to that of CG, with a further decrease as the disease progressed. This finding underscores the importance of pRNFL as an effective biomarker for assessing the severity of DR. The pRNFL consists of axons of retinal ganglion cells in the inner retinal layer, and its thickness reflects the health of the optic nerve fibers. In DR patients, microvascular damage and metabolic disorders induced by chronic hyperglycemia lead to retinal ischemia, increased inflammation, and subsequently affect the function and structural integrity of the nerve fibers [22,23]. Therefore, the reduction in pRNFL thickness is not only a sensitive indicator of DR progression but also an important basis for early diagnosis and intervention. Additionally, changes in increased optic disc area and CDR were observed, while the disc margin area decreased. These changes reveal the impact of NPDR on the optic nerve head structure, particularly the risk of optic nerve atrophy. As DR progresses, retinal nerve fibers are gradually damaged, and the tissue within the optic disc region may undergo remodeling, manifesting as an enlargement of the optic disc area, an increase in CDR, and a narrowing of the disc margin. These structural changes not only reflect the loss of optic nerve fibers but may also lead to insufficient blood flow perfusion around the optic disc, exacerbating the local hypoxic condition and further damaging optic nerve function [24,25]. Furthermore, it was found that the lower half of the ppVD was greatly reduced in NPDR patients, which may reflect the degeneration or loss of the vascular network beneath the optic disc. ppVD refers to the vascular density around the optic disc and serves as an important indicator of the microcirculatory health of this region [26]. A lower ppVD indicates a reduction in the blood supply to the retina through this area, which is crucial for maintaining the function of optic nerve fibers [27]. The decrease in ppVD is consistent with the reduction in pRNFL thickness, both pointing to damage to the optic nerve fibers [28]. The degeneration of the vascular network beneath the optic disc may be attributed to endothelial dysfunction under chronic hyperglycemic conditions, leading to poor angiogenesis, vascular sclerosis, and other issues, ultimately affecting the nutrient supply and oxygen delivery to the optic nerve fibers [29,30]. Although inconsiderable differences were found in optic cup volume across the groups, changes in optic disc area, CDR, and disc margin area provide additional information to aid in the identification of early DR presence and progression. The combined analysis of these parameters can more comprehensively describe the structural characteristics of the optic nerve head and its surrounding regions, helping clinicians better assess the progression of the disease and formulate personalized treatment plans [31,32].

This study used SS-OCTA technology to thoroughly analyze the changes in retinal structural parameters in the macular and optic disc regions of patients with NDR and different stages of NPDR. These findings enhance our understanding of the pathophysiological mechanisms of DR and provide valuable biomarkers for clinical diagnosis, aiding in the early identification and intervention of DR to protect patients' visual quality. However, this study has certain limitations. The retrospective design may lead to selection bias, as all included patients were from a single center and blood glucose control levels (such as HbA1c) or complications like diabetic nephropathy were not systematically recorded. Although the differences in disease duration between the groups were not statistically significant (*P* = 0.766), the average disease duration in the NPDR group (13.79 years) was longer than that in the NDR group (11.68 years), which may partly explain the deterioration of structural parameters. Future multicenter prospective studies are needed to control for confounding factors

and explore the dynamic associations between SS-OCTA parameters and metabolic indicators. Future work will focus on expanding sample sizes and promoting multicenter collaborations to enhance the reliability of results. Additionally, optimizing imaging technologies and analytical methods will ensure the consistency and accuracy of the data. It is also essential to explore the relationship between these parameters and other clinical factors, such as blood glucose control and disease duration, to establish a more comprehensive risk assessment model. Investigating the temporal changes of these parameters during DR progression will help evaluate treatment effects and guide the selection of personalized medical approaches. Furthermore, combining other imaging or biological markers to identify a comprehensive index system for early DR detection will be crucial. Ultimately, through continuous and in-depth research, we aim to better understand the mechanisms underlying DR development, improve diagnostic and therapeutic levels, and enhance the quality of life for patients.

## 5 Conclusion

In conclusion, relative to the healthy CG, both the NDR and NPDR groups showed drastic reductions in CVI, pRNFL, and blood flow density in the superficial and deep capillary plexuses, while the FAZ area and perimeter in the macula were notably enlarged. Additionally, changes in optic disc parameters, such as increased optic disc area and CDR, and reduced disc rim area, suggest that NPDR affects the structure of optic nerve head, highlighting the potential risk of optic nerve atrophy and its relevance for predicting visual function impairment. Despite limitations such as a small sample size and measurement precision constraints, the findings emphasize the significant value of SS-OCTA in detecting early retinal structural and functional changes associated with DR. Future research should further explore the relationships between these parameters and other clinical factors, optimize personalized treatment plans, and validate the results of this study to improve diagnostic and therapeutic approaches for DR patients, ultimately enhancing their quality of life.

## Supporting information

**S1 File. Data.**
(XLSX)

## Author contributions

**Conceptualization:** Jin Jiang, Xiaole Wang.

**Data curation:** Jin Jiang, Xiaole Wang.

**Formal analysis:** Jin Jiang, Hongjun Bian.

**Funding acquisition:** Jin Jiang.

**Investigation:** Jin Jiang, Xiaole Wang.

**Methodology:** Xiaole Wang, Hongjun Bian.

**Project administration:** Jin Jiang, Hongjun Bian.

**Resources:** Jin Jiang, Xiaole Wang.

**Software:** Xiaole Wang, Hongjun Bian.

**Supervision:** Jin Jiang, Xiaole Wang, Hongjun Bian.

**Validation:** Jin Jiang, Hongjun Bian.

**Writing – original draft:** Jin Jiang, Xiaole Wang, Hongjun Bian.

**Writing – review & editing:** Jin Jiang, Xiaole Wang, Hongjun Bian.

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
