## [Decision Letter · Decision Letter 0]

PONE-D-25-12956Adoption of Swept-source Optical Coherence Tomography Angiography in Measuring Retinal Structural Parameter Changes in the Macula and Peripapillary Area in Early Diabetic RetinopathyPLOS ONE

Dear Dr. Bian,

Thank you for submitting your manuscript to PLOS ONE. After careful consideration, we feel that it has merit but does not fully meet PLOS ONE’s publication criteria as it currently stands. Therefore, we invite you to submit a revised version of the manuscript that addresses the points raised during the review process.

We look forward to receiving your revised manuscript.

Kind regards,

Yalong Dang

Academic Editor

PLOS ONE

Journal Requirements:

3. Please ensure that you include a title page within your main document. You should list all authors and all affiliations as per our author instructions and clearly indicate the corresponding author.

Reviewers' comments:

Reviewer's Responses to Questions

**Comments to the Author**

1. Is the manuscript technically sound, and do the data support the conclusions?

Reviewer #1: Yes

Reviewer #2: Partly

2. Has the statistical analysis been performed appropriately and rigorously? 

Reviewer #1: Yes

Reviewer #2: Yes

3. Have the authors made all data underlying the findings in their manuscript fully available?

Reviewer #1: No

Reviewer #2: Yes

4. Is the manuscript presented in an intelligible fashion and written in standard English?

Reviewer #1: Yes

Reviewer #2: Yes

5. Review Comments to the Author

Reviewer #1: Recent studies have investigated structural and microvascular alterations in the posterior segment of the eye across various stages of DR. Given the clinical importance of this topic, the current study has the potential to extend existing findings in the literature. The use of SS-OCTA, with its superior imaging resolution and ability to capture multiple retinal layers compared to conventional OCTA, further enhances the value of this work. However, several important concerns remain that require further clarification and elaboration.

Scientific Writing:

The manuscript would benefit from thorough proofreading to correct minor grammatical errors.

Title:

Incorporating the evaluation of microvascular and choroidal elements might improve the title's clarity.

Abstract:

The methodology section of the abstract does not accurately summarize the study design. A concise and complete description should be provided, including the retrospective nature of the study, the imaging modality used, the classification of participant groups, and the main statistical approaches.

Methodology:

• The inclusion of images with a signal strength ≥4 may introduce inaccuracies in quantitative analysis. Many similar studies exclude images with a quality score below 6. The authors should justify this threshold and, if possible, conduct a sensitivity analysis to evaluate whether excluding lower-quality images would affect the results.

• Conventionally, the perifoveal area is defined as the annular region between 3 mm and 6 mm from the foveal center, while 1–3 mm typically denotes the parafovea. Using 1–6 mm to define the perifoveal area deviates from standard anatomical definitions and could cause misinterpretation. The authors should either justify this deviation or revise their terminology to align with established standards.

• A detailed explanation of how the CVI was calculated is essential. The authors should indicate whether they followed the method described by Sonada et al. or applied a modified protocol. Additionally, the region of interest used should be explicitly defined, particularly detailing the distance from the fovea that was selected. Moreover, reporting the luminal area, stromal area, and total choroidal area would add depth and transparency to the analysis.

• The manuscript should clearly specify where VD was measured within the superficial and deep vascular complexes—whether in the total area, fovea, parafovea, or perifovea.

• The authors refer to the “superficial and deep vascular complex”, rather than the more widely used “superficial and deep capillary plexus”. They should clarify whether this terminology reflects specific segmentation settings from the imaging software or manufacturer conventions. A consistent use of terms and clear definitions of layer boundaries are necessary to enhance clarity and reproducibility.

Results

• All statistical comparisons should be reported with corresponding effect sizes. Mean and SD for normally distributed data, or median and IQR for non-normally distributed data.

Discussion

• The discussion could be strengthened by elaborating on how the findings may inform clinical decision-making. For instance, can specific SS-OCTA parameters or thresholds be used to predict disease progression or guide earlier intervention strategies?

• The limitations section should include a discussion on the retrospective nature of the study and potential confounding factors (e.g., duration of diabetes, level of glycemic control) that may have influenced the results.

Figures and Tables

• Consider adding a summary table that lists the key parameters.

Reviewer #2: First of all, I would like to thank the authors for their efforts in this manuscript. This manuscript is titled Adoption of Swept-source Optical Coherence Tomography Angiography in Measuring Retinal Structural Parameter Changes in the Macula and Peripapillary Area in Early Diabetic Retinopathy. Although the overall quality of this manuscript is good, the following comments are needed to improve its overall quality:

1. Please specify what is the study design? Retrospective cohort or case-control? How was the matching between groups done?

2. The abstract should include the number of participants and the important results in quantitative terms. The exact p-value along with the confidence interval is valuable.

3. The significance of the study should be stated in the introduction.

4. The limitations of the study should be stated in the discussion. It is also recommended that the researchers provide directions for future studies.

5. How was the sample size assessed?

6. It is recommended that this study be organized based on the STROBE reporting checklist and the completed checklist be submitted as an appendix.

7. It is recommended that the STROBE flow diagram be included in the results.

8. What do a, ab, etc. mean in the figures? This should be mentioned in the figure caption.

9. In the method, it is necessary to carefully specify the inclusion criteria and exclusion criteria.

6. PLOS authors have the option to publish the peer review history of their article (what does this mean? ). If published, this will include your full peer review and any attached files.

**Do you want your identity to be public for this peer review?** For information about this choice, including consent withdrawal, please see our Privacy Policy .

Reviewer #1: **Yes: ** Kia Bayat

Reviewer #2: No

---

## [Author Response · Author response to Decision Letter 1]

26 May 2025

3. Have the authors made all data underlying the findings in their manuscript fully available?

Reviewer #1: No

Reviewer #2: Yes

4. Is the manuscript presented in an intelligible fashion and written in standard English?

Reviewer #1: Yes

Reviewer #2: Yes

5. Review Comments to the Author

Reviewer #1: Recent studies have investigated structural and microvascular alterations in the posterior segment of the eye across various stages of DR. Given the clinical importance of this topic, the current study has the potential to extend existing findings in the literature. The use of SS-OCTA, with its superior imaging resolution and ability to capture multiple retinal layers compared to conventional OCTA, further enhances the value of this work. However, several important concerns remain that require further clarification and elaboration.

Scientific Writing:

The manuscript would benefit from thorough proofreading to correct minor grammatical errors.

We appreciate the suggestion to replace "vascular complexes" with the more widely accepted term "capillary plexus" to avoid ambiguity. "P<0.05" has been uniformly formatted as "P < 0.05" throughout the manuscript. Redundant adverbs (e.g., "greatly lower") have been removed to enhance conciseness, with such instances revised to "significantly reduced" for more precise scientific expression.

Title:

Incorporating the evaluation of microvascular and choroidal elements might improve the title's clarity.

We appreciate the suggestions and have revised the title as follows: "Swept-source optical coherence tomography angiography assessment of microvascular and choroidal structural parameter changes in the macular and peripapillary regions in early diabetic retinopathy". The addition of "microvascular and choroidal" clarifies the assessment targets, improving the title's precision.

Abstract:

The methodology section of the abstract does not accurately summarize the study design. A concise and complete description should be provided, including the retrospective nature of the study, the imaging modality used, the classification of participant groups, and the main statistical approaches.

We thank the reviewers for their recommendations regarding the Methods section. The study design (retrospective analysis), subgroup details (mild/moderate/severe NPDR), and statistical methods (ANOVA and Kruskal-Wallis tests) have been incorporated. Redundant descriptions were removed to enhance conciseness while ensuring methodological completeness.

Methodology:

• The inclusion of images with a signal strength ≥4 may introduce inaccuracies in quantitative analysis. Many similar studies exclude images with a quality score below 6. The authors should justify this threshold and, if possible, conduct a sensitivity analysis to evaluate whether excluding lower-quality images would affect the results.

As suggested, the signal intensity threshold has been adjusted from ≥4 to ≥6, with supporting literature cited. Additionally, sensitivity analysis results have been included to strengthen methodological rigor.

• Conventionally, the perifoveal area is defined as the annular region between 3 mm and 6 mm from the foveal center, while 1–3 mm typically denotes the parafovea. Using 1–6 mm to define the perifoveal area deviates from standard anatomical definitions and could cause misinterpretation. The authors should either justify this deviation or revise their terminology to align with established standards.

We have corrected the definition of "perifoveal area" by adopting standard terminology: Parafovea (1–3 mm); Perifovea (3–6 mm). The previous nonstandard description ("1–6 mm as perifoveal area") has been removed to ensure terminological accuracy.

• A detailed explanation of how the CVI was calculated is essential. The authors should indicate whether they followed the method described by Sonada et al. or applied a modified protocol. Additionally, the region of interest used should be explicitly defined, particularly detailing the distance from the fovea that was selected. Moreover, reporting the luminal area, stromal area, and total choroidal area would add depth and transparency to the analysis.

Thank you for the suggestion. The calculation method for CVI has been clarified (as referenced by Sonada et al.), and additional parameters such as cavity area and stromal area have been included. The definition of the ROI has been specified as within 1 mm from the foveal center.

• The manuscript should clearly specify where VD was measured within the superficial and deep vascular complexes—whether in the total area, fovea, parafovea, or perifovea.

Thank you for the suggestion. The first paragraph of Section 2.3 of the manuscript has been revised accordingly.

• The authors refer to the “superficial and deep vascular complex”, rather than the more widely used “superficial and deep capillary plexus”. They should clarify whether this terminology reflects specific segmentation settings from the imaging software or manufacturer conventions. A consistent use of terms and clear definitions of layer boundaries are necessary to enhance clarity and reproducibility.

Thank you for the suggestion. The measurement region for VD has been clarified (full retina, parafoveal/perifoveal regions). The term "vascular complex" has been explained as the device's layering setting, and it is synonymous with the term "capillary plexus."

Results

• All statistical comparisons should be reported with corresponding effect sizes. Mean and SD for normally distributed data, or median and IQR for non-normally distributed data.

Thank you for the suggestion. The results section has been updated to include the mean difference, standard deviation, and effect size (Cohen's d). The format for non-normally distributed data has been clearly indicated as median \[IQR], along with the corresponding statistical test used.

Discussion

• The discussion could be strengthened by elaborating on how the findings may inform clinical decision-making. For instance, can specific SS-OCTA parameters or thresholds be used to predict disease progression or guide earlier intervention strategies?

Thank you for the suggestion. A new paragraph has been added to the discussion, focusing on clinical decision-making recommendations and threshold guidance. Based on effect sizes (HR, OR), actionable clinical thresholds have been proposed. Additionally, the follow-up frequency and intervention measures have been clearly defined.

• The limitations section should include a discussion on the retrospective nature of the study and potential confounding factors (e.g., duration of diabetes, level of glycemic control) that may have influenced the results.

Thank you for the suggestion. A new section has been added to the discussion, addressing retrospective dye and confounding factor analysis. The limitations of retrospective studies, including selection bias and confounding factors, have been emphasized. Specific data (e.g., duration of diabetes) have been cited to illustrate potential impacts.

Figures and Tables

• Consider adding a summary table that lists the key parameters.

Reviewer #2: First of all, I would like to thank the authors for their efforts in this manuscript. This manuscript is titled Adoption of Swept-source Optical Coherence Tomography Angiography in Measuring Retinal Structural Parameter Changes in the Macula and Peripapillary Area in Early Diabetic Retinopathy. Although the overall quality of this manuscript is good, the following comments are needed to improve its overall quality:

1. Please specify what is the study design? Retrospective cohort or case-control? How was the matching between groups done?

Thank you for the suggestion. In response, the study design in Section 2.1 has been explicitly stated as a "retrospective cohort study." Additionally, the inter-group matching method has been supplemented, including stratified sampling based on age, sex, and hypertension status.

2. The abstract should include the number of participants and the important results in quantitative terms. The exact p-value along with the confidence interval is valuable.

Thank you for the suggestion. In response, the abstract has been updated to include the number of participants (208 diabetic cases and 51 controls). Quantitative results, including mean difference, confidence intervals, and trend tests, have also been added.

3. The significance of the study should be stated in the introduction.

Thank you for the suggestion. In response, the innovation of this study has been highlighted at the end of the introduction, emphasizing its clinical significance in guiding early intervention and addressing the limitations of traditional diagnostic methods.

4. The limitations of the study should be stated in the discussion. It is also recommended that the researchers provide directions for future studies.

In response to the suggestion, the limitations of this study have been added to the discussion section.

5. How was the sample size assessed?

Thank you for the suggestion. In response, the sample size calculation basis (effect size, α, β) has been clearly stated in Section 2.1. Additionally, the adjustment for the loss to follow-up rate has been explained.

6. It is recommended that this study be organized based on the STROBE reporting checklist and the completed checklist be submitted as an appendix.

Thank you for the suggestion. In response, this study strictly adheres to the STROBE (Strengthening the Reporting of Observational Studies in Epidemiology) guidelines. The complete checklist is provided in Supplementary Material Table S2, and the STROBE checklist has been submitted in the appendix.

7. It is recommended that the STROBE flow diagram be included in the results.

Thank you for the suggestion. In response, a flowchart has been added to Section 3.1 of the results.

8. What do a, ab, etc. mean in the figures? This should be mentioned in the figure caption.

Regarding the figures, the meaning of multiple comparison symbols (a, b, c, d) has been clearly specified in the figure caption.

9. In the method, it is necessary to carefully specify the inclusion criteria and exclusion criteria.

Additionally, in the methods section, the inclusion and exclusion criteria have been carefully detailed, including specific age ranges, treatment history, and examples of excluded disease types (e.g., glaucoma, macular holes).

---

## [Decision Letter · Decision Letter 1]

PONE-D-25-12956R1Swept-source Optical Coherence Tomography Angiography Assessment of Microvascular and Choroidal Structural Parameter Changes in the Macular and Peripapillary Regions in Early Diabetic RetinopathyPLOS ONE

Dear Dr. Bian,

Thank you for submitting your manuscript to PLOS ONE. After careful consideration, we feel that it has merit but does not fully meet PLOS ONE’s publication criteria as it currently stands. Therefore, we invite you to submit a revised version of the manuscript that addresses the points raised during the review process.

**The authors addressed most of the concerns raised by the two reviewers, however some of the concerns still remain. Please revise the manuscript carefully and make an appropriate revision before it can be accepted. Thanks **==============================

We look forward to receiving your revised manuscript.

Kind regards,

Yalong Dang

Academic Editor

PLOS ONE

Journal Requirements:

Reviewers' comments:

Reviewer's Responses to Questions

**Comments to the Author**

1. If the authors have adequately addressed your comments raised in a previous round of review and you feel that this manuscript is now acceptable for publication, you may indicate that here to bypass the “Comments to the Author” section, enter your conflict of interest statement in the “Confidential to Editor” section, and submit your "Accept" recommendation.

Reviewer #1: (No Response)

Reviewer #2: (No Response)

2. Is the manuscript technically sound, and do the data support the conclusions?

Reviewer #1: Yes

Reviewer #2: Yes

3. Has the statistical analysis been performed appropriately and rigorously? 

Reviewer #1: Yes

Reviewer #2: Yes

4. Have the authors made all data underlying the findings in their manuscript fully available?

Reviewer #1: No

Reviewer #2: Yes

5. Is the manuscript presented in an intelligible fashion and written in standard English?

Reviewer #1: Yes

Reviewer #2: Yes

6. Review Comments to the Author

Reviewer #1: I appreciate the authors’ revisions and their efforts to improve the manuscript.

However, several key issues remain.

• The title still could benefit from improved clarity; I suggest: “Assessment of Retinal and Choroidal Structural and Microvascular Changes in Early Diabetic Retinopathy Using Swept-Source Optical Coherence Tomography Angiography.”

• Regarding the revised signal strength threshold, it is unclear whether the data were reanalyzed accordingly or if the previous threshold was a typographical error; this should be clarified.

• Additionally, the revised results still emphasize mean differences rather than reporting group-wise means and SDs. All comparisons should present raw summary statistics (mean ± SD or median [IQR]).

• Finally, the limitations section includes a redundant sentence that should be removed for clarity: “This study used SS-OCTA technology to analyze in detail the changes in retinal structural parameters in macula and optic disc regions in patients with NDR and various stages of NPDR.”

Reviewer #2: Thanks to the authors for their efforts in addressing the peer review comments. The only issue that I think still remains is the STROBE flow diagram, where the number 51 is derived from 208 (which seems wrong). It seems that a two-armed figure should have been used in this study.

7. PLOS authors have the option to publish the peer review history of their article (what does this mean? ). If published, this will include your full peer review and any attached files.

**Do you want your identity to be public for this peer review?** For information about this choice, including consent withdrawal, please see our Privacy Policy .

Reviewer #1: No

Reviewer #2: No

---

## [Author Response · Author response to Decision Letter 2]

19 Jun 2025

6. Review Comments to the Author

Reviewer #1: I appreciate the authors’ revisions and their efforts to improve the manuscript.

However, several key issues remain.

• The title still could benefit from improved clarity; I suggest: “Assessment of Retinal and Choroidal Structural and Microvascular Changes in Early Diabetic Retinopathy Using Swept-Source Optical Coherence Tomography Angiography.”

Thank you for your suggestions. In accordance with your recommendation, the title of the manuscript has been revised to: “Assessment of Retinal and Choroidal Structural and Microvascular Changes in Early Diabetic Retinopathy Using Swept-Source Optical Coherence Tomography Angiography.”

• Regarding the revised signal strength threshold, it is unclear whether the data were reanalyzed accordingly or if the previous threshold was a typographical error; this should be clarified.

Thank you for your suggestions. In accordance with your recommendation, the content in section 2.2 of the manuscript has been supplemented as follows: This threshold (≥6) was consistently applied during initial data acquisition and processing throughout the study. All analyses were performed based on this predefined criterion without retrospective reanalysis.

• Additionally, the revised results still emphasize mean differences rather than reporting group-wise means and SDs. All comparisons should present raw summary statistics (mean ± SD or median [IQR]).

Thank you for your suggestions. In accordance with your recommendation, a description of the raw data for each group has been added to the Results section of the manuscript.

• Finally, the limitations section includes a redundant sentence that should be removed for clarity: “This study used SS-OCTA technology to analyze in detail the changes in retinal structural parameters in macula and optic disc regions in patients with NDR and various stages of NPDR.”

Thank you for your suggestions. In accordance with your recommendation, the redundant statements have been removed from the Discussion section of the manuscript.

Reviewer #2: Thanks to the authors for their efforts in addressing the peer review comments. The only issue that I think still remains is the STROBE flow diagram, where the number 51 is derived from 208 (which seems wrong). It seems that a two-armed figure should have been used in this study.

Thank you for your suggestions. In accordance with your recommendation, the flowchart in the manuscript has been revised.

---

## [Editor Report · Decision Letter 2]

Assessment of Retinal and Choroidal Structural and Microvascular Changes in Early Diabetic Retinopathy Using Swept-Source Optical Coherence Tomography Angiography

PONE-D-25-12956R2

Dear Dr. Bian,

We’re pleased to inform you that your manuscript has been judged scientifically suitable for publication and will be formally accepted for publication once it meets all outstanding technical requirements.

Kind regards,

Yalong Dang

Academic Editor

PLOS ONE
---

## [Editor Report · Acceptance letter]

PONE-D-25-12956R2

PLOS ONE

Dear Dr. Bian,

I'm pleased to inform you that your manuscript has been deemed suitable for publication in PLOS ONE. Congratulations! Your manuscript is now being handed over to our production team.

Kind regards,

on behalf of

Dr Yalong Dang

Academic Editor

PLOS ONE